# Improvement of Cecal Commensal Microbiome Following the Insect Additive into Chicken Diet

**DOI:** 10.3390/ani10040577

**Published:** 2020-03-30

**Authors:** Agata Józefiak, Abdelbasset Benzertiha, Bartosz Kierończyk, Anna Łukomska, Izabela Wesołowska, Mateusz Rawski

**Affiliations:** 1Department of Preclinical Sciences and Infectious Diseases, Poznań University of Life Sciences, Wołyńska 35, 60–637 Poznań, Poland; anna.lukomska@up.poznan.pl (A.Ł.); izabela.wesolowska@up.poznan.pl (I.W.); 2Department of Animal Nutrition, Poznań University of Life Sciences, Wołyńska 33, 60-637 Poznań, Poland; abdelbasset.benzertiha@hipromine.com (A.B.); bartosz.kieronczyk@up.poznan.pl (B.K.); 3HiProMine S.A., Poznańska 8, 62-023 Robakowo, Poland; 4Division of Inland Fisheries and Aquaculture, Institute of Zoology, Poznań University of Life Sciences, Wojska Polskiego 71c, 60-625 Poznań, Poland; mateusz.rawski@up.poznan.pl

**Keywords:** chicken, cecum, microbiome, GIT, insect diet, probiotic bacteria

## Abstract

**Simple Summary:**

Throughout their lifecycle, insects can be a rich source of many valuable nutrients and biologically active components. It was demonstrated that the additive of *Tenebrio molitor* (TM) and *Zophobas morio* (ZM) into broiler chicken diet can have a positive effect on their growth parameters as well as some microbial composition in the gastrointestinal tract (GIT). Therefore, the present study evaluated the effect of diet with the addition of *Tenebrio molitor* and *Zophobas morio* (0.2 and 0.3%) on the cecal microbiome in broilers. The addition of 0.2% ZM compared to the negative control (NC) group resulted in an increase in relative abundance of the Actinobacteria, including the family Bifidobacteriaceae, with the highest relative abundance of genus *Bifidobacterium pseudolongum*. The addition of 0.2% ZM resulted in an increase in the number of *Lactobacillus agilis*. The highest relative abundance of the family Ruminococcaceae was observed in the 0.2 TM group together with *Lactobacillus reuteri* but with no significant differences. Furthermore, a significant level of Clostridia was observed in the 0.2 TM group. A relatively small addition of *Tenebrio molitor* and *Zophobas morio* to broiler diet can modulate commensal and probiotic microbiome composition in the cecum and increase the relative abundance of positive bacteria to have a positive impact on gut health of broilers.

**Abstract:**

Gastrointestinal microbiota play an important role in regulating the metabolic processes of animals and humans. A properly balanced cecal microbiota modulates growth parameters and the risk of infections. The study examined the effect of the addition of 0.2% and 0.3% of *Tenebrio molitor* and *Zophobas morio* on cecal microbiome of broilers. The material was the cecum digesta. The obtained DNA was analyzed using 16S rRNA next generation sequencing. The results of the study show that the addition of a relatively small amount of *Z*. *morio* and *T*. *molitor* modulates the broiler cecum microbiome composition. The most positive effect on cecal microbiota was recorded in the 0.2% *Z*. *morio* diet. A significant increase in the relative amount of genus *Lactobacillus*, represented by the species *Lactobacillus agilis* and the amount of bacteria in the Clostridia class, was observed. Moreover, the addition of 0.2% ZM resulted in a significant increase of relative abundance of the family Bifidobacteriaceae with the highest relative abundance of genus *Bifidobacterium pseudolongum*. The obtained results indicate that the addition of a relatively small amount of insect meal in broiler diet stimulates colonization by probiotic and commensal bacteria, which may act as barriers against infection by pathogenic bacteria.

## 1. Introduction

The flora of the gastrointestinal tract (GIT) of poultry is a complex bacterial ecosystem, with the last part of the GI tract being the most densely populated. The following factors have the biggest impact on species diversity and the distribution of microorganisms in the digestive system: pH, aerobic conditions, and availability of nutrients [1]. Bacteria are involved in the regulation of metabolic and immunological processes [2]. They have a significant effect on growth indicators and immunity to infections caused by potentially pathogenic microorganisms such as: *Salmonella*, *Campylobacter*, *Escherichia,* or *Enterococcus*. By modulating gastrointestinal microbiome with the use of natural nutritional factors, it is possible to improve health and breeding parameters of poultry [3,4]. Recent years have seen major developments in the area of research on inducing changes in the composition of the intestinal ecosystem with the use of feed additives in the form of pre- and probiotics as well as insects as an alternative source of protein and antimicrobial peptides [5,6].

Modern tools of molecular biology enable precise identification and taxonomic classification of the microbiome. The application of metagenomic techniques based on 16S rRNA sequencing enables the analysis of the entire bacterial population in a specific environment. This method consists of amplifying the bacterial DNA and creating libraries that are then marked and sequenced. The applied next generation sequencing (NGS) technique allows one to compare the obtained bacterial DNA sequences with selected taxonomic bases and to analyze the entire microbiome present in a given material in one study without the need to conduct separate microbiological analyses. Currently, high-throughput next generation sequencing (HT-NGS) is applied using various systems, including: 454 Roche, Illumina, or SOLiD [7]. The analysis of a microbiome using the genome sequencer with Miseq’s Illumina platform is based on the analysis of the 16S rRNA bacterial gene amplicons. This procedure requires prior in vitro amplification of the genetic material using polymerase chain reaction (PCR).

The term “microbiome” refers to the collective bacteria and other microorganisms in an ecosystem [8]. In broiler chickens, the most abundant microorganisms belong to the Firmicutes, the Proteobacteria, and the Bacteroidetes phyla [8,9]. Less numerous are the representatives of Actinobacteria, Tenericutes, Cyanobacteria, and Fusobacteria [10,11,12]. The upper parts of the GI tract, such as the crop, the proventriculus, and the gizzard, are populated with lactic acid bacteria, *Lactobacillus* spp., which participate in the decomposition of starch and lactate fermentation processes and are capable of producing bacteriocins [13,14]. The gizzard is adapted to mechanically grinding up food. Low pH value in this section of the GI tract is correlated with a smaller population of microorganisms [13]. The analysis of the lower parts of the digestive tract show that the small intestine is populated by the representatives of *Lactobacillus*, *Enterococcus,* and *Clostridiaceae*, whereas the large intestine is a more complex bacterial ecosystem [8,15]. One of the parts of the GI tract that plays an important role in the avian digestive system is the cecum, consisting of two diverticula, where key processes of water absorption regulation and fermentation of indigestible carbohydrates occur [11]. This section of the avian GI tract is populated by bacteria that belong to the Firmicutes, the Bacteroides, and the Proteobacteria phyla [13]. In the Firmicutes phylum, Clostridia are the most numerous, with clearly marked Clostridiaceae, Lachnospiraceae, and Ruminococcaceae families [10,16,17]. The representatives of Enteroccaceae, Enterobacteriaceae, and Bacteroidaceae families are also present but less common [18]. The following species of microorganisms are commonly represented in the cecum: *Bacteroides fragilis*, *Lactobacillus crispatus*, *Lactobacillus johnsonie*, *Lactobacillus reuteri,* and *Lactobacillus salivarius* [10,19]. Densely populated with microorganisms, the cecum plays an important role in the distribution of dietary fiber to short-chain fatty acids (SCFA). As a result, more energy is supplied to the host, which positively correlates with the improvement of growth indicators in a number of animal species, including poultry [8].

The formation of a protective intestinal barrier initiates the first contact with the external environment. One of the most important factors affecting the microbiota in different sections of the GI tract is nutrition. It has been established that age, genotype, and stress factors, such as housing conditions and stock density, affect the formation of the bacterial ecosystem of the GI tract of poultry [10,20].

Due to the high content of fat and amino acids, insects can be a suitable source of nutrients for animals in livestock production [21,22]. In addition, insects were found to be one of the nutrient factors, and biological active components can have beneficial effects on gastrointestinal tract health to modulate the microbiome diversity [23,24,25]. This microbiome modulation can be achieved through such compounds as chitin, lipids, and antimicrobial peptides (AMPs). The aim of the study was to analyze the effect of insect diet based on yellow mealworm (*Tenebrio molitor*) and super mealworm larvae (*Zophobas morio*) on the cecal digesta microbiome diversity of broilers chicken.

## 2. Materials and Methods

According to Polish law and Directive 2010/63/EU of the European Parliament and of the Council of 22 September 2010 on the protection of animals used for scientific purposes, the experiment conducted as part of the study does not require the approval of the Local Ethical Committee for Animal Experiments in Poznań.

### 2.1. Birds and Housing

In the experiment, 600 one-day-old broilers (ROSS 308) randomly assigned to 6 experimental groups were used. Each test was carried out in 10 repetitions with 10 birds per replication. The animals were kept in 1.00 × 1.00 m chicken coops (Piast, Olszowa Experimental Unit, no. 0161, Olszowa, Poland) with 23 h light access for the first week and 19 h light access from days 7 to 21. From days 22 to 35, the lighting parameters were similar to those used for the first week of production of hens for fattening. Vaccination against Gumboro disease was done for all birds at day 21 (AviPro PRECISE, Lohmann Animal GmbH, Cuxhaven, Germany).

### 2.2. Diets and Feeding Program

The chicks had unlimited access to feed for 35 days. The feed was prepared in a loose form. The raw materials in the feed mix were ground using a disc mill (Skiold A/S, Saby, Denmark) with a 2.5 mm gap between the discs and then subjected to the necessary heat treatment processes. The feed was produced in the Piast Pasze feed mill (Lewkowiec, Poland) in compliance with ISO 9001:2008. The ingredients and the calculated nutritive value of the basal diet are presented in Table 1.

From days 1 to 14, the birds were fed a starter mix, and from days 15 to 35, a grower mix. In the negative control group (NC), the basic diet without feed additives was applied, while in the positive control group (PC), the basic feed with salinomycin (60 mg/kg of feed) was applied. In other dietary groups, the feed was enriched with insect meal according to the following experimental system: 0.2% *Tenebrio molitor* (TM02), 0.2% *Zophobas morio* (ZM02), 0.3% *Tenebrio molitor* (TM03), 0.3% *Zophobas morio* (ZM03). Full-fat insect meal was added on top.

*Tenebrio molitor* and *Zophobas morio* larvae were obtained from a commercial source (HiProMine S.A., Robakowo, Poland). Insects were dried in a laboratory dryer (SLN 240, POL-EKO Aparatura, Poland) for 24 h at 50 °C and then ground (Zelmer, 1900 w, Rzeszów, Poland).

### 2.3. Sampling

At the end of the experiment, 1 bird was randomly selected as a repetition. The animals were killed by manual cervical dislocation. Then, the cecum was dissected in order to obtain its contents. The portion of cecal samples was gently squeezed from 1 bird per pen and next pooled by 2 birds per sample (5 replicates of digesta; *n* = 5), and immediately packed, sealed in sterilized plastic bags, frozen, and stored at −80 °C for analyses of the microbial populations by next-generation sequencing (NGS).

### 2.4. Bacterial DNA Extraction and Amplification

DNA was extracted with a commercial kit (Sherlock AX, A&A Biotechnology, Poland) according to the manufacturer’s instructions. Samples were mechanically lysed on FastPrep - 24 on Zirconia beads (A&A Biotechnology, Poland) and additionally lysed enzymatic towards bacteria. The presence of bacterial DNA in the samples was confirmed using Real-Time PCR on termocycler Mx3000P (Stratagene, USA) with SYBR Green as fluorochrome. In the reaction for amplification of 16S rDNA, the following universal reaction primers were used: 1055F 5′-ATGGCTGTCGTCAGCT-3′ and 1392R 5′-ACGGGCGGTGTGTAC-3′. The temperature profile of reaction was: 95 °C, 3 min; 95 °C, 15 s; 58 °C, 30 s; 72 °C, 30 s; Tm 65 °C -> 95 °C.

### 2.5. Cecal Digesta Microbiome 16SrRNA Sequencing

DNA was quantified using the NanoDrop and standardized at 5 ng/μl. Microbial diversity was studied by sequencing the amplified V3-V4 region of the 16S rRNA gene by using primers 16S Amplicon PCR Forward Primer 5′ TCGTCGGCAGCGTCAGATGTGTATAAGAGACAGCCTACG GGNGGCWGCAG 16S Amplicon PCR Reverse Primer 5′ GTCTCGTGGGCTCGGAGATGTGTATA AGAGACAGGACTACHVGGGTATCTAATCC. PCR conditions: 95 °C for 3 min; 25 cycles of: 95 °C for 30 s, 55 °C for 30 s, 72 °C for 30 s, 72 °C for 5 min, hold at 4 °C. The expected size on a Bioanalyzer trace after the Amplicon PCR step is ~550 bp. The PCR products were cleaned up step uses AMPure XP beads. The libraries were sequenced running 2 × 300 bp paired-end reads. The PCR products were cleaned, and the library was combined with the sequencing adapters and the dual indices using the Nextera XT Index Kit (Illumina, San Diego, CA, USA), according to 16S Metagenomic Sequencing Library Preparation instruction (Illumina, San Diego, USA). The PCR with Nextera XT Index Primers was carry out in following conditions: 95 °C for 3 min; 8 cycles of 95 °C for 30 s, 55 °C for 30 s, 72 °C for 30 s, 72 °C for 5 min, hold at 4 °C. The PCR products were cleaned up again with AMPure XP beads. The library was validated to the expected size on a Bioanalyzertrace for the final library of ~630 bp. The libraries were quantified using a fluorometric quantification method using dsDNA binding dyes. Individual concentrations of DNA libraries were calculated in nM, based on the size of DNA amplicons, as determined by an Agilent Technologies 2100 Bioanalyzer.

For sequencing, the individual libraries were diluted for 4 nM, denaturated with 10 mM Tris pH 8.5, and spiked with 20% (*v*/*v*) of PhiX. Aliquots with 5 μl of diluted DNA were mixed for pooling libraries preparation for MiSeq ((Illumina, San Diego, CA, USA) run. Then, >100,000 reads were performed per sample.

### 2.6. Metagenomic Analysis

The microbiome sequences were classified according the V3 and the V4 amplicons and analyzed using a database of 16S rRNA data. Specific sequences, 341F and 785R, were used for the amplification and the libraries preparation. For the table PCR reaction with Q5 Hot Start High-Fidelity 2X Master Mix available, reaction conditions were performed in accordance with the manufacturer’s requirements. Sequencing took place on the MiSeq sequencer in paired-end (PE) technology at 2 × 250 nt using Illumina v2 kit. Automatic initial data analysis was performed on the MiSeq apparatus using the MiSeq Reporter (MSR) v2.6 software. The analysis consisted of two stages: automatic demultiplexing of samples and generating fastq files containing raw reads. The output of sequencing was a classification of reads at several taxonomic levels: kingdom, phylum, class, order, family, genus, and species. Quality analysis of the sequence was conducted with quality control and filtration to obtain high-quality sequences. Valid sequences were screened from samples according to the barcode at both ends of the sequence and were corrected for the direction by the primer sequences. All valid and filtered sequences were clustered into operational taxonomic units (OTUs) based on a 97% 16S rRNA gene sequence identity level. The obtained sequences were checked with BLAST (Basic Local Alignment Search Tool) [26] and searched against the Greengenes database (http://greengenes.lbl.gov) to determine the phylogeny of the OTU. The results were classified at several taxonomic levels: kingdom, phylum, class, order, family, genus, and species. Relative abundance profiles of cecal microbiota were established according to OTU abundance of different groups.

Bioinformatic analysis ensuring the classification of readings by species level was carried out with the QIIME software package based on the GreenGenes v13_8 reference sequence database [27,28]. The analysis consisted of stages: 1) removal of adapter sequences—cutadapt program; 2) quality analysis of readings and removal of low-quality sequences (quality <20, minimum length 30)—cutadapt program [29]; 3) paired sequence connection—fastq-join algorithm (http://code.google.com/p/ea-utils); 4) clustering based on the selected base of reference sequence—the uclust algorithm [30]; 5) chimer removal sequence—ChimeraSlayer algorithm [31]; 6) assigning taxonomy to a selected base of reference sequences—the uclust algorithm [27,30].

### 2.7. Statistical Analysis

The experiments had a completely randomized design. The sample obtained by pooling digesta from 2 birds was defined as the experimental unit, i.e., *n* = 5 per treatment (10 birds per group, 1 randomly chosen chick from each pen, 2 birds pooled per sample).

Bioinformatic analysis was carried out using the R program and using phyloseq, vegan, and factoextra packages, while charts were generated using the gglpot2 and the ggbiplot packages [32].

Kruskal–Wallis H tests were used to assess whether the values originated from the same distribution or whether their distribution was different depending on the group they belonged to.

The beta diversity measure was calculated based on the Bray–Curtis method [33]. Selected sequences representing OTU were compared to the Genbank database using the BLAST algorithm [26].

The relative abundance of data were tested for normal distributions using the Kolmogorow–Smirnov test. An analysis of variance was conducted using Bartlett’s test. The RMSE (root square error of the mean) was calculated. The RMSE equation is: *RMSE = √MSE*, where MSE is mean square error. The significance of differences among groups was determined with Duncan’s multiple range test at the significance level of *p* < 0.05. The analyses were performed using SAS software (SAS Institute Inc., Cary, NC, USA).

The following general model was used:Y_i_ = μ + α_i_ + δ_ij_,
where Y_i_ is the observed dependent variable, μ is the overall mean, α_i_ is the effect of insects’ meal, and δ_ij_ is the random error. The RMSE equation is: *RMSE = √MSE*, where MSE is mean square error, ^a-c^ means within a row with no common superscripts differ significantly (*p ≤* 0.05).

## 3. Results

The Illumina MiSeq was performed using 30 samples to generate a total 2,161,838 raw sequence reads. After passing the quality filter, there were 2,132,133 (98.63%) sequences. A relative abundance of bacteria was recorded in all experimental groups (99.37–99.77%).

### 3.1. Effect on Cecal Comensal Microbiome

Consideration of alpha diversity within the sequence datasets using the number of observed OTUs, Chao1, ACE, Shannon and Simpson indices, showed no significant variation associated with 0.2% *Tenebrio molitor*, 0.2% *Zophobas morio*, 0.3% *Tenebrio molitor*, and 0.3% *Zophobas morio* (ZM03) (Table 2, Figure 1). The beta diversity among samples is presented on plots of individual samples in Figure 2. Clusters were superimposed over the PCA analysis and represent the differences within a group of samples. The results from the statistical analysis of the obtained sequences for most prevalence commensal bacterial are presented in Table 3.

We found that cecal digesta microbiome differed between researched groups (PC, NC, TM02, ZM02, TM02 and ZM03). The relative abundance of Actinobacteria and Clostridia showed statistically significant differences (*p* < 0.05). The statistical differences of relative abundance in the dietary groups under study were also confirmed on the family level in Bifidobacteriaceae and Ruminococcaceae. The 16SrRNA OUTs analysis at the level of species beside Bifidobacterium (*p* = 0.021) also showed statistical differences in relative abundance of *Lactobacillus agilis (p <* 0.001).

#### 3.1.1. Phylum Level

The results of OUTs analysis of the obtained 16S rDNA sequences for the taxonomic classification of bacterial phyla are presented in Figure 1. Analysis of bacterial diversity in cecal digesta indicated more than 40 different phylas. In the studies conducted at the level of phylum, a reduction of the relative number of Actinobacteria was observed in TM02, TM03, and ZM03 (Table 1), while in the PC group and the 0.2% ZM group, in comparison to the NC group, an increase of the relative number of Actinobacteria was observed with statistically significant differences (*p* = 0.039) (Table 1). There was a visible decrease in the relative number of Firmicutes in all of the studied dietary groups in comparison with the negative control group. The same downward trend was observed in the positive control group compared to the negative control. However, in the phylum Firmicutes, the observed changes could not be confirmed statistically (*p* = 0.131) (Figure 3). The addition of the *Tenebrio molitor* meal to the diet caused a decrease in the relative number of Proteobacteria. In TM02, TM03, ZM03, and PC groups, a higher relative number of Bacteroidetes was observed in comparison to the negative control group. No changes in the Bacteroidetes population were observed in the cecum contents of chickens fed on a diet with 0.2% addition of the *Zophobas morio* meal. The changes in the relative abundance of Proteobacteria and Bacteroidetes did not show statistical differences (*p* > 0.05).

#### 3.1.2. Class Level

In the analysis of the obtained 16S rDNA sequences within the taxonomic rank of class, a decrease in Clostridia was observed in the groups that were fed on a diet with the addition of salinomycin *Zophobas morio* and 0.3% TM *(p =* 0.032). An increase in the relative number of the Clostridia was observed in the 0.2% TM group *(p =* 0.032) (Figure 4). The addition of *Zophobas morio* caused an increase in the relative number of Bacilli, with a similar effect observed in the cecum contents of chickens for fattening in the salinomycin group, but observed changes were not statistically confirmed (*p >* 0.05). Dietary supplementation with *Tenebrio molitor* showed a decrease in the amount of Actinobacteria with respect to the positive control group (*p* = 0.021) (Table 1). With regard to Actinobacteria, an increase in their relative number in the ZM02 group and a more visible effect in the positive control group were observed (*p* = 0.021).

In the diet groups TM02, TM03 and ZM03, an increase of relative abundance of Bacteroidia was observed with no statistical differences. These bacteria remained at a similar level in the ZM02 and the NC groups (*p* > 0.05).

#### 3.1.3. Order Level

The OTU analysis identified 42 different orders of bacteria in cecum digesta of broilers. The addition of insect meal caused a decrease in the relative number of Clostridiales in groups ZM02 and ZM03 in comparison with the negative control group, and the results were similar to the positive control group (*p* = 0.032) (Table 3). A slight increase in Clostridiales was observed in the TM02 group, but increasing the amount of *Tenebrio molitor* to 0.3% resulted in a decrease in the amount of Clostridiales (*p* = 0.032). The addition of *Zophobas morio* to the diet resulted in an increase of the relative number of Lactobacillales in relation to the control group at a level similar to that observed in the salinomycin group. The diet with the addition of 0.2% *Tenebrio molitor* caused a decrease in the relative number of Lactobacillales, whereas an upward trend was observed for the diet with the addition of 0.3% *Tenebrio molitor* but with no statistical differences. The addition of 0.2% *Zophobas morio* caused an increase of Bifidobacteriales (Table 3). The highest increase in Bifidobacteriales in the cecum contents of broiler chickens was observed in the PC and the ZM02 groups with statistically significant differences (*p* = 0.021).

### 3.2. Family

The results of the OTU analysis showed 73 different bacterial families. The study showed that the feed additives used reduced the relative abundance of microorganisms from the Ruminococcaceae family—with the exception of 0.2% *Tenebrio molitor* supplementation—compared to the negative control group (*p* = 0.021). A lower number of Lachnospiraceae was observed in comparison with the negative control group, while the addition of salinomycin caused an increase in the number of bacteria from the analyzed family. In the ZM and the PC groups, a similar increase in the number of Lactobacillaceae was observed. When applied, TM showed a decrease in the number of Enterobacteriaceae and Bifidobacteriaceae (*p* = 0.021). However, the addition of 0.2% ZM resulted in an increase in the relative number of bacteria belonging to family Enterobacteriaceae and Bifidobacteriaceae. With the exception of chickens fed on a diet with the addition of 0.2% *Zophobas morio* meal, an increase in the number of Bacteroidaceae was observed in comparison to the results obtained from chickens fed on a basic diet without feed additives.

### 3.3. Genus

The taxonomical analysis of the obtained 16S rRNA sequences indicated more than 100 different bacterial genera. Supplementation with 0.3% TM stimulated the growth of *Ruminococcus* in comparison to other dietary groups. In comparison with the negative control group, decreases of *Faecalibacterium* and *Blautia* were observed, with the exception of a diet with the addition of 0.3% TM, but were not significantly affected (Table 3). The *Bacteroides* and the *Bifidobacterium* populations changed in a similar fashion as in the cases of Bacteroidaceae and Bifidobacteriaceae. An increase of *Bifidobacterium* was observed as a result of both TM diets. In the case of diets ZM02 or 03, a decrease in *Bifidobacterium* relative abundance was observed. The effect of insect-based diets on cecal microbiota in broilers translated to a relative abundance of *Bifidobacterium* and *Ruminococcus*. A decrease in the *Lactobacillus* population was observed in the group supplemented with 0.2% TM, whereas the addition of ZM caused an increase of microorganisms at a similar level as in the positive control group, but the relative abundance of genus *Lactobacillus* was also affected significantly.

### 3.4. Species

A significant increase of *Lactobacillus agilis* was observed in the case of a relatively moderate supplementation with *Zophobas morio* (0.2%) with high statistical differences (*p* < 0.001) (Table 3). The highest relative abundance of *Lactobacillus agilis* and an almost 10-fold increase was observed in the ZM 02 group (*p* < 0.001), whereas the addition of 0.2% TM stimulated the development of *Lactobacillus reuteri* more effectively than in the case of the positive control group but with no statistical differences (*p* > 0.05). In general, all dietary groups showed lower levels of *Faecalibacterium prausnitzii* in comparison with the results obtained from chickens fed on a basic diet without feed additives. Moreover, an increase in the relative number of *Bifidobacterium pseudolongum* was observed in 0.2% ZM and PC groups.

## 4. Discussion

Gastrointestinal microorganisms play an important role in the body’s defense mechanisms, regulating the immune system and acting as a barrier to pathogenic microorganisms. Bacteria influence health and growth parameters in animal breeding. By maintaining a balanced GI tract microbiome, it is possible to stimulate the growth of commensal bacteria and inhibit the growth of potentially pathogenic bacteria. Establishing a proper balance within the intestinal microbiome is a very important factor in improving animal health. Due to the high content of nutrients, proteins, and peptides modulating gastrointestinal microbial activity, insects could be a valuable addition to poultry nutrition.

This study of alpha and beta diversity confirmed high individual variability between chickens previously described by Singh et al. (2014) [34]. It has been shown that the cecum is naturally populated with Firmicutes, Actinobacteria, Proteobacteria, and Bacteroidetes. The results obtained are in accordance with the available literature and confirm studies conducted by other authors [10,13,35]. Moreover, the studies on chicken broilers showed a decrease in the number of microorganisms from the Firmicutes phylum in all of the studied dietary groups. The addition of the *Tenebrio molitor* meal caused smaller changes in the Firmicutes population in the cecum content than in the case of the addition of *Zophobas morio*. According to the available literature, the results of the quantitative analysis of the Firmicutes population show a predisposition to inflammation and intestinal cancer. It is generally believed that higher levels of Firmicutes increase the risk of diseases, while a decrease in the total number of Firmicutes is positively correlated with intestinal health [36]. There is little literature available on the influence of the *Tenebrio molitor* meal on gastrointestinal microbiota in broiler chickens. However, other results were obtained by other authors, such as Biasato et al. 2018 [37], who analyzed the impact of the addition of 7.5% *Tenebrio molitor* meal to the diet of free-range chickens and observed an increase in the relative number of Firmicutes. When analyzing Firmicutes, it is very important to take into account the ratio of *Bacteroides* to *Lactobacilli* population [36]. A decrease in the amount of *Bacteroides* and an increase in *Lactobacilli* are said to be positively correlated with intestinal health. In the conducted studies, the increase in the population of *Lactobacilli* and the decrease in *Bacteroides* were most clearly visible when using the addition of 0.2% *Zophobas morio*.

Clostridia, classified in the order Clostridiales and involved in the metabolism of short-chain fatty acids (SCFA), were the most numerous in the class rank [13]. These results confirm a previous study conducted by Gong et al. (2007) [16]. The obtained data indicate that, in the cecum, the class of Clostridiales is most often represented by Lachnospiraceae and Ruminococcaceae with a smaller population of Enterobacteriaceae and Bacteroidaceae also present [16]. Similar results were obtained by Park et al. (2016) [16] when observing a small population of Bifidibacteriaceae.

The studies showed that the addition of 0.2 and 0.3% of *Zophobas morio* to the diet of broiler chickens causes an increase in the relative number of *Lactobacillus* bacteria. The addition of 0.2% ZM compared to the NC group resulted in a significant increase in the relative abundance of Actinobacteria, including the family Bifidobacteriaceae with the highest relative abundance of genus *Bifidobacterium pseudolongum*. Bifidobacteria are generally considered to promote intestinal health by restricting intestinal colonization by pathogens. They lower intestinal pH increases fermentation and may boost immunity effects. As documented by Vasquez et al. (2009) [38], *Bifidobacterium pseudolongum* exerted an effect on oxidative stress and protected the intestine through a relative predominance of protective species. Furthermore, other studies showed that *B*. *pseudolongum* exhibit an inhibitory activity against *Salmonella* Typhi N15 and *Escherichia coli*—EHEC (enterohaemorragic *Escherichia coli*) [39]. The results of our study indicate that the addition of 0.2% ZM resulted in a significant increase of *Lactobacillus agilis*. Lactobacilli produce lactic acid and proteolytic enzymes. It was indicated that *Lactobacillus agilis* is a probiotic bacteria that may be used in livestock production to enhance nutrient digestion and modulate animal growth through feed conversion efficiency and health parameters by limiting pathogen infection levels.

Similar results were obtained by Wei et al. 2013 [40]. The addition of 0.3% of *Tenebrio molitor* stimulates the increase of the relative number of *Ruminococcus* bacteria. The obtained results are consistent with the study published by Bisato et al. (2018) [37]. *Ruminococcus* bacteria demonstrate the ability to ferment carbohydrates with simultaneous production of SCFAs and constitute a rich group of probiotic bacteria [37]. It can be suggested that a low concentration of insect meal supports the development of beneficial flora.

## 5. Conclusions

A relatively small amount of *Z*. *morio* and *T*. *molitor* (0.2 and 0.3%) added to the complete diet of broiler chickens can improve their cecal commensal microbiome. The statistical differences were not observed in the whole metagenomic data. We indicated that the addition of *Z*. *morio* resulted in an increase of the relative abundance of Actinobacteria, including family Bifidobacteriaceae, and the addition of *T*. *molitor* resulted in a significant increase of the relative abundance of family Ruminococcaceae. The obtained results indicate that the addition of insect meal in broiler diets stimulates cecal colonization by probiotic and commensal bacteria to develop barriers against infection with pathogenic bacteria.

## Figures and Tables

**Figure 1 animals-10-00577-f001:**
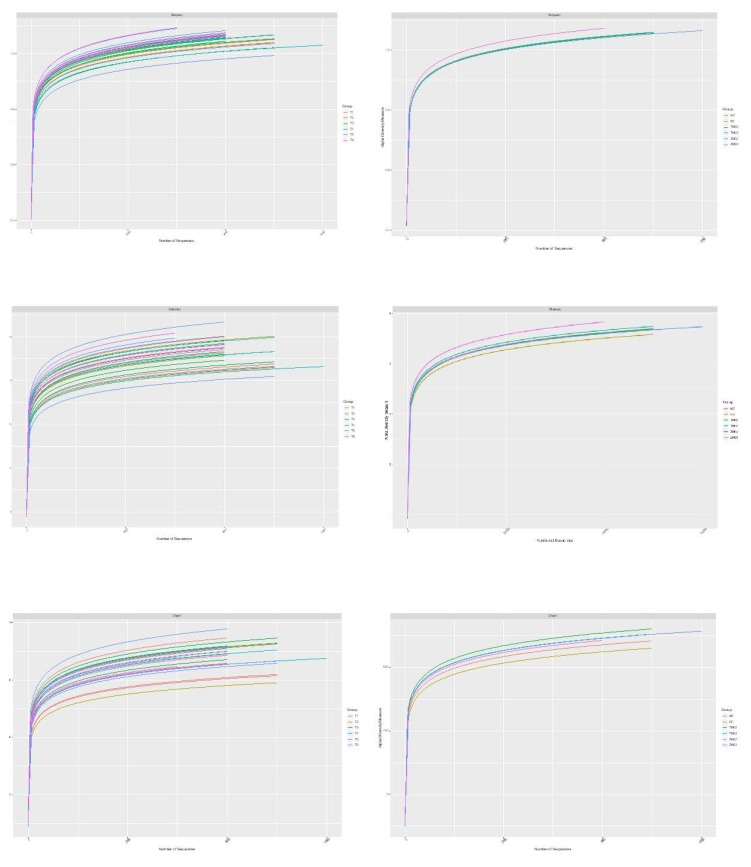
Rarefaction curves of alpha diversity cecum digesta, legends refer to sample: PC—positive control (salinomycin, 60 ppm); NC—negative control (no additives); TM02—(0.2% *T. molitor* full-fat meal); ZM02—(0.2% *Z. morio* full-fat meal); TM03—(0.3% *T. molitor* full-fat meal); ZM03—(0.3% *Z. morio* full-fat meal).

**Figure 2 animals-10-00577-f002:**
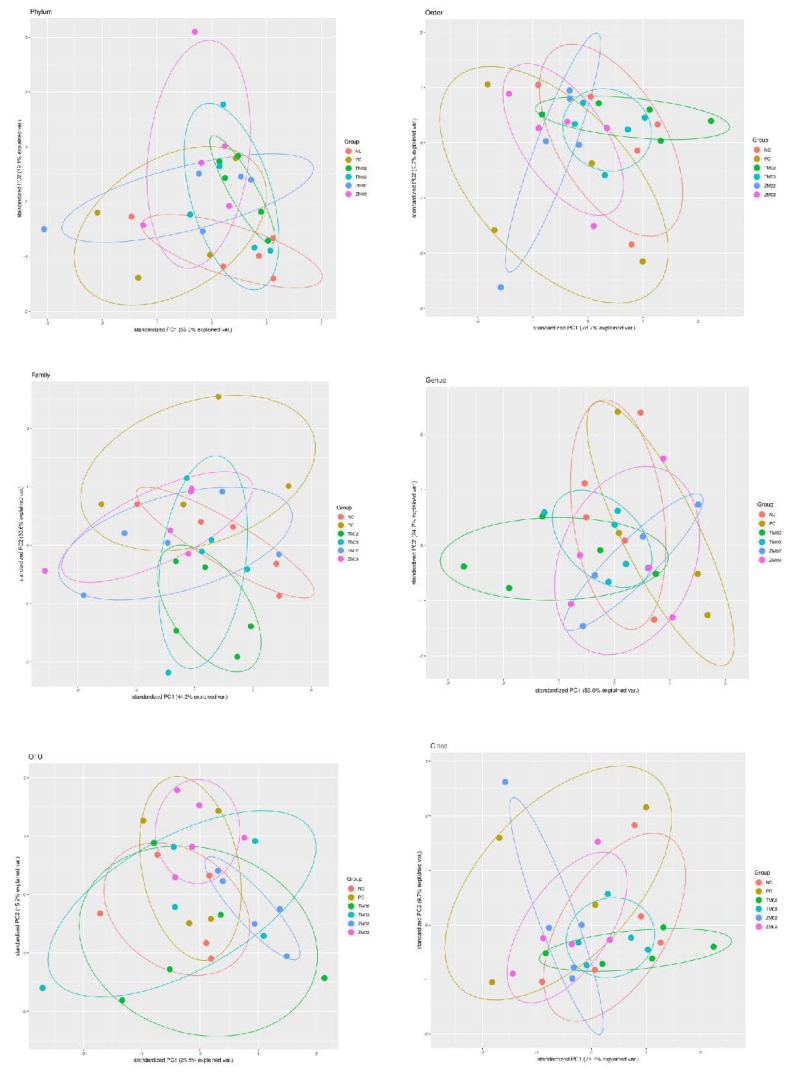
Principal component analysis (PCA) of obtained sequence from cecal digesta samples: PC—positive control (salinomycin, 60 ppm); NC—negative control (no additives); TM02—(0.2% *T. molitor* full-fat meal); ZM02—(0.2% *Z. morio* full-fat meal); TM03—(0.3% *T. molitor* full-fat meal); ZM03—(0.3% *Z. morio* full-fat meal).

**Figure 3 animals-10-00577-f003:**
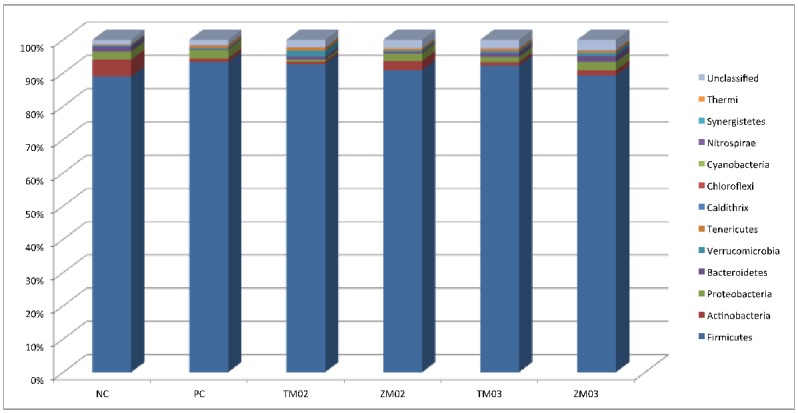
Effect of insect diet on chicken cecal microbiome composition at phylum-level. Legends refer to sample: PC—positive control (salinomycin, 60 ppm); NC—negative control (no additives); TM02—(0.2% *T. molitor* full-fat meal); ZM02—(0.2% *Z. morio* full-fat meal); TM03—(0.3% *T. molitor* full-fat meal); ZM03—(0.3% *Z. morio* full-fat meal).

**Figure 4 animals-10-00577-f004:**
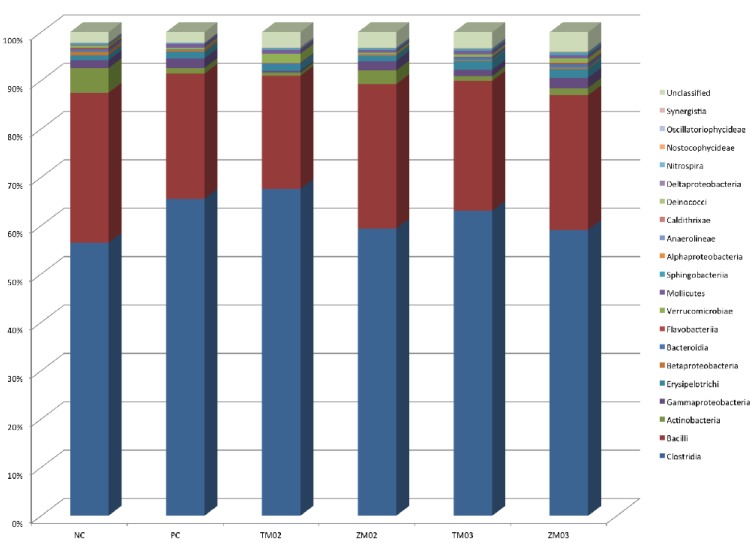
Effect of insect diet on chicken cecal microbiome composition at class-level. Legends refer to sample: PC—positive control (salinomycin, 60 ppm); NC—negative control (no additives); TM02—(0.2% *T. molitor* full-fat meal); ZM02—(0.2% *Z. morio* full-fat meal); TM03—(0.3% *T. molitor* full-fat meal); ZM03—(0.3% *Z. morio* full-fat meal).

**Table 1 animals-10-00577-t001:** Composition of the experimental basal diets.

Ingredients (g·kg^−1^)	1–14 d	15–35 d
Wheat	487.4	513.4
Rye	100.0	100.0
Soybean meal	207.8	169.5
Rapeseed meal	100.0	100.0
Fish meal	20.0	20.0
Soybean oil	49.9	71.1
Vitamin–mineral premix ^a^	3.0	3.0
Monocalcium phosphate	13.1	6.7
Limestone	8.0	6.8
Salt (NaCl)	1.1	1.3
Sodium carbonate (Na_2_CO_3_)	2.2	1.7
L–Lysine HCl	2.9	2.4
Methionine 88% liquid	3.1	2.5
L–Threonine	1.5	1.6
Titanium dioxide (TiO_2_) ^b^	-	2.0
Calculated nutritive value (g·kg^−1^)		
Crude protein	215.6	200.6
Ether extract	65.4	86.3
Crude fiber	33.1	32.2
Total phosphorus (P)	7.9	6.3
Calcium (Ca)	8.5	7.0
Methionine	6.1	5.3
Lysine	12.5	11.2
Methionine + cysteine	9.9	9.0
Threonine	9.1	8.6
AME_N_ (MJ·kg^−1^)	12.56	13.31

^a^ Provided the following per kilogram of diet: vitamin A, 11,166 IU; cholecalciferol, 2,500 IU; vitamin E, 80 mg; menadione, 2.50 mg; vitamin B_12_, 0.02 mg; folic acid, 1.17 mg; choline, 379 mg; D–pantothenic acid, 12.50 mg; riboflavin, 7.0 mg; niacin, 41.67 mg; thiamine, 2.17 mg; D–biotin, 0.18 mg; pyridoxine, 4.0 mg; ethoxyquin, 0.09 mg; Mn (MnO_2_), 73 mg; Zn (ZnO), 55 mg; Fe (FeSO_4_), 45 mg; Cu (CuSO_4_), 20 mg; I (CaI_2_O_6_), 0.62 mg; and Se (Na_2_SeO_3_), 0.3 mg. ^b^ Replaced the corresponding amount of wheat in each diet from 30 to 35 d of broiler growth.

**Table 2 animals-10-00577-t002:** Results of alpha diversity comparison among groups by Kruskal–Walls test.

	PC	NC	TM02	ZM02	TM03	ZM03	Chi-Squared	*df*	*p*-Value
Chao	1353,433689	1403,584812	1460,172589	1482,72019	1473,615167	1440,336954	2.5531	5	0.7715
inverse Simpson	24,04508234	26,76290487	20,86621696	27,18980723	29,13428671	29,73953594	4.4531	5	0.4862
OTU	1002,75	1048,6	1113,8	1099,6	1098	1066,2	3.5919	5	0.6095
Shannon	4,188089975	4,282324207	4,251156956	4,298738827	4,338615676	4,403900425	2.7703	5	0.7353
Simpson	0,957585469	0,960598925	0,949845404	0,960809514	0,955005403	0,965599882	4.4531	5	0.4862

PC—positive control (salinomycin, 60 ppm); NC—negative control (no additives); TM02—(0.2% *T*. *molitor* full-fat meal); ZM02—(0.2% *Z*. *morio* full-fat meal); TM03—(0.3% *T*. *molitor* full-fat meal); ZM03—(0.3% *Z*. *morio* full-fat meal); (*df*)—degrees of freedom; OUT—operational taxonomic units.

**Table 3 animals-10-00577-t003:** Relative abundance of bacterial communities in cecal digesta of chickens fed: 0.2% *Tenebrio molitor* (TM02), 0.2% *Zophobas morio* (ZM02), 0.3% *Tenebrio molitor* (TM03), 0.3% *Zophobas morio* (ZM03).

	Treatment												RMSE	*p*-Value
	PC	SD	NC	SD	TM02	SD	ZM02	SD	TM03	SD	ZM03	SD		
Kingdom														
Bacteria	99.77	0.09	99.41	0.16	99.37	0.33	99.43	0.20	99.53	0.23	99.53	0.15	0.002	*0.070*
Unclasified	0.23	0.09	0.59	0.16	0.63	0.33	0.57	0.20	0.47	0.23	0.47	0.15	0.002	*0.070*
Phylum														
Firmicutes	89.47	3.07	93.42	3.67	92.49	1.99	89.85	3.40	92.14	2.77	89.51	2.72	0.028	*0.131*
Actinobacteria	4.89^a^	0.00	1.66^b^	0.00	1.19^b^	0.00	3.58^ab^	0.00	1.37^b^	0.00	1.52^b^	0.00	0.020	*0.039*
Bacteroidetes	1.42	2.09	0.12	0.14	0.62	1.05	0.12	0.29	0.84	1.33	1.44	2.17	0.014	*0.535*
Proteobacteria	2.30	3.36	2.34	2.34	0.32	0.28	2.18	3.08	1.36	1.03	2.25	2.56	0.024	*0.730*
Class														
Actinobacteria	4.33^a^	0,00	0.87^b^	0,00	0.25^b^	0,00	2.05^ab^	0,00	0.69^b^	0,00	0.87^b^	0,00	0.019	*0.021*
Bacilli	25.60	5.89	21.40	6.25	18.71	5.62	26.09	0.76	20.85	4.61	24.24	4.50	0.053	*0.210*
Clostridia	62.79^b^	7.15	70.70^ab^	6.24	72.66^a^	7.19	62.69^b^	3.76	69.42^ab^	3.73	63.48^b^	4.64	0.058	*0.032*
Order														
Bifidobacteriales	4.32^a^	3.89	0.86^b^	166	0.24^b^	0.34	2.04^ab^	2.32	0.67^b^	1.05	0.86^b^	1.16	0.019	*0.021*
Lactobacillales	25.49	5.78	21.30	6.14	18.51	5.61	25.79	0.81	20.67	4.64	24.12	4.47	0.052	*0.209*
Clostridiales	62.79^b^	7.03	70.70^ab^	5.86	72.66^a^	6.99	62.69^b^	4.05	69.42^ab^	3.18	63.48^b^	4.86	0.058	*0.032*
Family														
Bifidobacteriaceae	4.32^a^	3.89	0.86^b^	1.66	0.24^b^	0.34	2.04^ab^	2.32	0.67^b^	1.05	0.86^b^	1.16	0.019	*0.021*
Lactobacillaceae	24.57	6.84	20.56	5.68	18.12	5.70	24.64	1.37	20.26	4.89	23.84	4.45	0.054	*0.314*
Lachnospiraceae	25.85	5.26	23.13	1.88	17.16	2.55	17.51	5.57	20.91	5.08	20.95	6.95	0.054	*0.136*
Ruminococcaceae	21.90^b^	5.39	30.54^ab^	7.95	33.48^a^	4.11	25.24^ab^	6.15	29.53^ab^	5.09	21.83^b^	6.18	0.065	*0.042*
Genus														
*Bifidobacterium*	4.32^a^	3.89	0.86^b^	1.66	0.24^b^	0.34	2.04^ab^	2.32	0.67^b^	1.05	0.86^b^	1.16	0.019	*0.021*
*Lactobacillus*	24.57	6.78	20.56	5.61	18.12	5.65	24.64	1.38	20.26	4.82	23.84	4.42	0.054	*0.314*
*Blautia*	4.10	4.48	4.65	1.70	4.14	3.14	4.39	4.98	4.97	4.27	3.54	5.38	0.027	*0.973*
*Ruminococcus*	14.49^c^	1.90	20.73^abc^	2.64	26.57^a^	1.57	18.41^bc^	1.41	22.85^ab^	1.71	15.21^c^	2.64	0.054	*0.012*
*Faecalibacterium*	5.39	6.27	7.97	8.52	5.22	4.36	4.81	5.03	4.59	4.91	4.95	6.49	0.034	*0.640*
Species														
*Bifidobacterium pseudolongum*	4.32^a^	3.82	0.86^b^	1.56	0.24^b^	0.29	2.04^ab^	2.28	0.67^b^	0.95	0.86^b^	1.03	0.019	*0.021*
*Lactobacillus agilis*	1.99^b^	1.51	1.63^b^	2.47	0.99^b^	1.17	10.62^a^	3.15	1.29^b^	1.13	1.14^b^	0.68	0.019	*<0.001*
*Lactobacillus reuteri*	10.14	4.11	8.26	5.13	10.97	3.61	8.35	2.66	8.75	4.66	7.29	2.14	0.038	*0.669*
*Faecalibacterium* *prausnitzii*	5.39	3.67	7.97	4.20	5.22	2.50	4.81	3.44	4.59	2.70	4.95	3.89	0.034	*0.640*

Means represent 10 birds in 5 pooled samples, 2 birds per sample (*n* = 5); PC—positive control (salinomycin, 60 ppm); NC—negative control (no additives); TM02—(0.2% *T*. *molitor* full-fat meal); ZM02—(0.2% *Z*. *morio* full-fat meal); TM03—(0.3% *T*. *molitor* full-fat meal); ZM03—(0.3% *Z*. *morio* full-fat meal); SD – standard deviation, RMSE—root square error of the mean. The RMSE equation is: *RMSE = √MSE*, where MSE is mean square error, ^a-c^ means within a row with no common superscripts differ significantly (*p ≤* 0.05).

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
