# Peer review of "Improvement of Cecal Commensal Microbiome Following the Insect Additive into Chicken Diet"

_animals, 2020, doi:10.3390/ani10040577_

Round 1

Reviewer 1 Report

The manuscript deals with the study on the effect of insect diet based on Tenebrio molitor) and Zophobas morio larvae on the cecal digesta microbiome diversity of broilers chicken. The research was performed with enough precision, but a minor revision is required to improve the paper.

In the following is a point-by-point list of possible changes:

  1. Simple Summary:

line 18: Add letters, to use later Tenebrio molitor (TM) Zophobas morio (ZM)

line 21: Add - negative control group (NC)

line 27: Tenebrio molitor

  1. Abstract:

Line 35: At the beginning write Tenebrio molitor and Zophobas morio, a later only T. molitor and Z. morio

Correct Line 42 ceca… cecal

  1. Materials and Methods:

Line 128: Correct 0,3% Tenebrio molitor (TM03)

Line 130: Add Zophobas morio (Fabr.) larvae and Tenebrio molitor L. imagines

  1. Results

Line 254: Add. (Fig…..)

Unify Figure or Fig.

Line 280: ZM 02% or 03% or both

Line 261 TM02% or 03% or both

  1. Discussion:

First literature is given as numbers and now as names. Please unify

  1. References:

Line 366 218 bold

369 217 bold

390 2014 bold

441 2016 bold

  1. Figure 1. The scale should be the same in each figure in the manuscript from 0-100% or 83-100%
  2. Figure 2. Taxonomy classes should be on vertical or horizontal axes and the same during the whole manuscript.

Author Response

We thank the Reviewer for the interest in our work and for their helpful comments, which will greatly improve the manuscript. We have done our best to respond to all the points raised in the review. Below the Reviewer will find our answers to the comments.

  1. Simple Summary:

line 18: Add letters, to use later Tenebrio molitor (TM) Zophobas morio (ZM)

line 21: Add - negative control group (NC)

line 27: Tenebrio molitor

It has been corrected in the text.

  1. Abstract:

Line 35: At the beginning write Tenebrio molitor and Zophobas morio, a later only T. molitor and Z. morio

Correct Line 42 ceca… cecal

It has been corrected in the text.

  1. Materials and Methods:

Line 128: Correct 0,3% Tenebrio molitor (TM03)

Line 130: Add Zophobas morio (Fabr.) larvae and Tenebrio molitor L. Imagines

It has been corrected in the text

  1. Results

Line 254: Add. (Fig…..)

Unify Figure or Fig.

The use of Fig has been unified to Figure.

Line 280: ZM 02% or 03% or both

It has been corrected on “both TM diet”.

Line 261 TM02% or 03% or both

It has been corrected on “ZM02% or 03% diet”

  1. Discussion:

First literature is given as numbers and now as names. Please unify

We unified literature as numbers according Animals guidelines for Authors

  1. References:

Line 366 218 bold

369 217 bold

390 2014 bold

441 2016 bold

Literature data were performed with Mendeley according References for Animals, but in some references were still mistakes. The years published data has been manually bolded and some spaces deleted.

Figure 1. The scale should be the same in each figure in the manuscript from 0-100% or 83-100%

The scale has unified in each figure 0-100%.

Figure 2. Taxonomy classes should be on vertical or horizontal axes and the same during the whole manuscript.

Figure has been changed on layout as on whole manuscript.

Best regards,

Agata Józefiak

Reviewer 2 Report

The introduction should contain more information about insects as a feed source. 

The metagenomics methods section does not provide sufficient details about the bioinformatics methods. Similarly, there are not sufficient details about the statistical comparisons or the methods used to compare microbial abundance data. The authors must start with large scale analysis including a PCA based on established methods such as Bray-Curtis distances. Also not sure why alpha and beta diversity indices are not reported. 

The results are difficult to judge without additional details about bioinformatics and statistical methods. 

Author Response

Response to Reviewer 2 Comments

We thank the Editor and Reviewer for their interest in our work and for their helpful comments which will greatly improve the manuscript. We have done our best to respond to all the points raised in the reviews. The Reviewer have raised certain points and we appreciate the opportunity to clarify our research objectives and results. As indicated below, we have checked all the general and specific comments pointed out line by line by the Reviewer and have made the necessary changes accordingly to their indications.

  1. It is not clear how many broilers were used for the microbiota analysis. 1 bird / replicate? 

Answer:

The Authors want to thank the Reviewer for this comment. By mistake the Authors forgot to add information about statistical model. The following paragraph was added to the Material and Methods section”

“2.7. Statistical Analysis

The experiments had a completely randomized design. The sample obtained by pooling digesta from 2 birds was defined as experimental unit, i.e., n=5 per treatment (10 birds per group, 1 randomly chosen chick from each pen, 2 birds pooled per sample). All data were tested for normal distributions using the Kolmogorow-Smirnov test. An analysis of variance was conducted using Bartlett’s test. The significance of differences among groups was determined with Duncan’s multiple range test at the significance level of p < 0.05. The analyses were performed using SAS software (SAS Institute Inc., Cary, NC, USA).

The following general model was used:

Yi=μ+αiij,

where Yi is the observed dependent variable, μ is the overall mean, αi is the effect of fat, and δij is the random error.”

„The portion of caecal samples was gently squeezed from 1 bird per pen and next pooled by 2 birds per sample (5 replicates of digesta; n=5), and immediately packed, sealed in sterilized plastic bags, frozen, and stored at -80℃ for analyses of the microbial populations by next-generation sequencing (NGS)."

  1. How many clean reads were obtained and analyzed?

Answer:

The following information was added in Results section:

“The Illumina MiSeq was performed using 30 samples to generate a total 2,161,838 raw sequence reads. After passing the quality filter in has been received 2,132,133 (98,63%) sequences.”

The Authors below present the numbers of obtained reads.

Sample

Raw read pairs

Passing QC

% passing QC

PC – 60

86383

85238

98,67%

PC – 61

84797

83425

98,38%

PC – 62

78450

77093

98,27%

PC – 63

70416

69616

98,86%

PC – 64

66279

65504

98,83%

NC – 65

66918

66172

98,89%

NC – 66

75790

74836

98,74%

NC – 67

72460

71630

98,85%

NC – 68

79169

78107

98,66%

NC – 69

74995

73668

98,23%

TM02 – 70

60714

59657

98,26%

TM02 – 71

66746

65950

98,81%

TM02 – 72

82943

81657

98,45%

TM02 – 73

62602

61899

98,88%

TM02 – 74

72809

71956

98,83%

ZM02 – 75

73625

72643

98,67%

ZM02 – 76

73820

72991

98,88%

ZM02 – 77

86383

85238

98,67%

ZM02 – 78

70988

70201

98,89%

ZM02 – 79

75793

74786

98,67%

TM03 – 80

76246

74822

98,13%

TM03 – 81

66490

65360

98,30%

TM03 – 82

79685

78742

98,82%

TM03 – 83

69755

68947

98,84%

TM03 – 84

59835

59079

98,74%

ZM03 – 85

67026

66192

98,76%

ZM03 – 86

67664

66872

98,83%

ZM03 – 87

55716

54952

98,63%

ZM03 – 88

73065

71717

98,16%

ZM03 – 89

64276

63183

98,30%

  1. Did the authors performed rarefaction analysis on the OTU table? This is important since difference in number of sequences among samples can affect the final results.

Answer:

All data were presented according results of OTU’s analysis.

Table 2.
Here there is the big problem: In the fist part of the table the authors says that 99% of the sequences belonging to bacteria, going at family or genus level the sum of the OTU does not reach 99%. For example at species level the sum of the OTU is around 21%. Where are the rest of the sequences?

We would like to present below the results of OUT’s analysis paired to bacterial sequences. The NGS metagenomic analysis of chickens’ fecal sample gives very high differences between samples (Singh et al., 2014). In the table 2 Authors presented only the most abundant commensal bacteria species confirmed by BLAST analysis and be interesting from the point of view of chicken health and breeding.

Singh, K.M., Shah, T.M., Reddy, B. et al. Taxonomic and gene-centric metagenomics of the fecal microbiome of low and high feed conversion ratio (FCR) broilers. J Appl Genetics 55, 145–154 (2014). https://doi.org/10.1007/s13353-013-0179-4

In addition the description of order, class and phylum shoulb be shortened because to my opinion the most interesting is the highest taxonomic resolution.

We agree with the Reviewer, however we would like to leave the description of order, class and phylum. Very high individual diversity between chickens microbiome on species and genus level incline to analysis of upper taxonomical levels be relevant to chickens broilers health.

In addition it is very difficult to find the species level of the LAB like L. agilis and L. reuteri. Did you double check the sequences by manually blast?

It is also not clear how the statistical test was performed.

Answer:

Statistical tests were described in Answer to Comments 1

We did BLAST analysis. The results for Lactobacillus agilis and Lactobacillus reuteri are present below.

BLAST Lactobacillus agilis,

301883

Description

Max Score

Total Score

Query Cover

E value

Per. Ident

Accession

Lactobacillus agilis strain ClaCZ26

784

784

100%

0

99.77%

MN055936.1

Lactobacillus agilis strain Yanat1

784

784

100%

0

99.77%

MG966464.1

Lactobacillus agilis strain IMAU50369

784

784

100%

0

99.77%

MG547310.1

Lactobacillus agilis strain IMAU50278

784

784

100%

0

99.77%

MF623242.1

Lactobacillus agilis strain HBUAS53196

784

784

100%

0

99.77%

MH393056.1

Lactobacillus agilis strain HBUAS53184

784

784

100%

0

99.77%

MH393044.1

322737

Uncultured bacterium, clone FS1075

761

761

100%

0

98.83%

FN667127.1

Lactobacillus agilis strain DSM 102821

756

756

100%

0

98.59%

MN537457.1

Lactobacillus agilis strain HBUAS54266

756

756

100%

0

98.59%

MH817717.1

756

756

100%

0

98.59%

MH393042.1

Lactobacillus agilis strain HBUAS53144

756

756

100%

0

98.59%

MH393004.1

Lactobacillus sp. strain UVAS:RY13

756

756

100%

0

98.59%

MG938657.1

316438

Lactobacillus agilis strain DSM 102821

462

462

100%

3E-126

99.60%

MN537457.1

Lactobacillus agilis strain HBUAS54266

462

462

100%

3E-126

99.60%

MH817717.1

Lactobacillus agilis strain HBUAS53182

462

462

100%

3E-126

99.60%

MH393042.1

Lactobacillus agilis strain HBUAS53144

462

462

100%

3E-126

99.60%

MH393004.1

Lactobacillus sp. strain UVAS:RY13

462

462

100%

3E-126

99.60%

MG938657.1

Lactobacillus sp. strain UVAS:RY11

462

462

100%

3E-126

99.60%

MG938655.1

Description

Max Score

Total Score

Query Cover

E value

Per. Ident

Accession

332919

Lactobacillus agilis strain ClaCZ26

466

466

100%

2E-127

100.00%

MN055936.1

Lactobacillus agilis strain Yanat1

466

466

100%

2E-127

100.00%

MG966464.1

Lactobacillus agilis strain IMAU50369

466

466

100%

2E-127

100.00%

MG547310.1

Lactobacillus agilis strain IMAU50278

466

466

100%

2E-127

100.00%

MF623242.1

Lactobacillus agilis strain HBUAS53196

466

466

100%

2E-127

100.00%

MH393056.1

Lactobacillus agilis strain HBUAS53184

466

466

100%

2E-127

100.00%

MH393044.1

356089

Lactobacillus agilis strain ClaCZ26

789

789

100%

0

100.00%

MN055936.1

Lactobacillus agilis strain Yanat1

789

789

100%

0

100.00%

MG966464.1

Lactobacillus agilis strain IMAU50369

789

789

100%

0

100.00%

MG547310.1

Lactobacillus agilis strain IMAU50278

789

789

100%

0

100.00%

MF623242.1

Lactobacillus agilis strain HBUAS53196

789

789

100%

0

100.00%

MH393056.1

Lactobacillus agilis strain HBUAS53184

789

789

100%

0

100.00%

MH393044.1

New.ReferenceOTU71

Lactobacillus agilis strain DSM 102821

776

776

100%

0

99.53%

MN537457.1

Lactobacillus agilis strain HBUAS54266

776

776

100%

0

99.53%

MH817717.1

Lactobacillus agilis strain HBUAS53182

776

776

100%

0

99.53%

MH393042.1

Lactobacillus agilis strain HBUAS53144

776

776

100%

0

99.53%

MH393004.1

Lactobacillus sp. strain UVAS:RY13

776

776

100%

0

99.53%

MG938657.1

Lactobacillus sp. strain UVAS:RY11

776

776

100%

0

99.53%

MG938655.1

New.CleanUp.ReferenceOTU31670

Lactobacillus agilis strain ClaCZ26

750

750

100%

0

98.36%

MN055936.1

Lactobacillus agilis strain Yanat1

750

750

100%

0

98.36%

MG966464.1

Lactobacillus agilis strain IMAU50369

750

750

100%

0

98.36%

MG547310.1

Lactobacillus agilis strain IMAU50278

750

750

100%

0

98.36%

MF623242.1

Lactobacillus agilis strain HBUAS53196

750

750

100%

0

98.36%

MH393056.1

Lactobacillus agilis strain HBUAS53184

750

750

100%

0

98.36%

MH393044.1

BLAST analysis Lactobacillus reuteri

Description

Max Score

Total Score

Query Cover

E value

Per. Ident

Accession

Lactobacillus reuteri strain YSJL-12 chromosome, complete genome

481

2848

100%

8E-132

100.00%

CP030089.1

Lactobacillus reuteri strain CE3

481

481

100%

8E-132

100.00%

MK920158.1

Lactobacillus reuteri strain WHH1689 chromosome, complete genome

481

2843

100%

8E-132

100.00%

CP027805.1

Uncultured bacterium clone OTU184

481

481

100%

8E-132

100.00%

MH222004.1

Uncultured bacterium clone 737

481

481

100%

8E-132

100.00%

MG716292.1

Uncultured bacterium clone 112

481

481

100%

8E-132

100.00%

KX593246.1

354971

Lactobacillus reuteri strain RR17

438

438

99%

5E-119

98.39%

MF093235.1

Description

Max Score

Total Score

Query Cover

E value

Per. Ident

Accession

Lactobacillus reuteri strain MM45

438

438

99%

5E-119

98.39%

MF093232.1

Lactobacillus reuteri strain BX54

438

438

99%

5E-119

98.39%

MF093230.1

Lactobacillus reuteri strain BX46

438

438

99%

5E-119

98.39%

MF093229.1

Lactobacillus reuteri strain BX19

438

438

99%

5E-119

98.39%

MF093227.1

Lactobacillus reuteri strain BR6

438

438

99%

5E-119

98.39%

MF093225.1

354256

Lactobacillus reuteri strain HM22

747

747

98%

0

98.58%

MK038963.1

Lactobacillus reuteri strain WHH1689 chromosome, complete genome

747

4433

99%

0

98.35%

CP027805.1

New.CleanUp.ReferenceOTU41758

Uncultured bacterium, clone: TTFN031

728

728

100%

0

97.42%

AB934502.1

Lactobacillus reuteri strain ClaCZ17

723

723

100%

0

97.19%

MN055933.1

Lactobacillus reuteri strain RRJ-33

723

723

100%

0

97.19%

MK572789.1

Lactobacillus reuteri strain RRJ-25

723

723

100%

0

97.19%

MK572787.1

Lactobacillus reuteri strain YSJL-12 chromosome, complete genome

723

4278

100%

0

97.19%

CP030089.1

Lactobacillus reuteri strain ABRIIN28

723

723

100%

0

97.19%

MG547724.1

I'm suggesting to performed a Principal Coordinate analysis on the beta diversity calculation in order to see difference among treatment, a statistical test should be also performed on the alpha diversity parameters.

We present the results of the alpha diversity and PC analysis on the beta diversity calculation.

In table 2 we present the statistical analysis of the most abundance bacteria and important according chickens breeding.

The histograms should be replaced by a table where the standard deviation should be displayed.

The results of standard deviation was added in table 2. Authors would like to leave the histograms of order, class, phylum and family level because the changes are very important for growth parameters and chicken health. In Authors opinion is easier to catch the differences on histograms. The results on histograms are also presented by other   Authors    

References:

  1. Park, S.H.; Lee, S.I.; Kim, S.A.; Christensen, K.; Ricke, S.C. Comparison of antibiotic supplementation versus a yeast-based prebiotic on the cecal microbiome of commercial broilers. PLOS ONE 2017, 12, e0182805.
  2. Park, S.H.; Lee, S.I.; Ricke, S.C. Microbial populations in naked neck chicken ceca raised on pasture flock fed with commercial yeast cell wall prebiotics via an Illumina MiSeq platform. PLoS ONE 2016, 11, e0151944.
  3. Singh, K.M.; Shah, T.M.; Reddy, B.; Deshpande, S.; Rank, D.N.; Joshi, C.G. Taxonomic and gene-centric metagenomics of the fecal microbiome of low and high feed conversion ratio (FCR) broilers. Journal of Applied Genetics 2014, 55, 145–154.

[1][2][3]

Best regards,

Agata Józefiak

Reviewer 3 Report

The findings are interesting and the sample size is excellent. However, the results and methods sections need work.

It is not clear how you determined the statistical significance of differences between the microbiomes. No mention of your statistical methods is made in the methods. A section 2.7 or another paragraph for section 2.6 is needed for this.

Table 2 mentions the RMSE, but this abbreviation is not defined elsewhere. What software produced this, and what does it mean?

Figure 1 appears to be cutoff: the Y-axis starts at 82%. I also don't know if the bar graph figures are useful. They are all different sizes, arranged differently, with missing data labels or cut-off labels, etc. Plus, we don't really need information on all these families, especially since the discussion seems to focus exclusively on Lactobacilli, Bacteroides, and a few other species.

An NMDS plot might be a better way to represent the differences in microbiome visually, else Table 2 is enough. Or, focus solely on the microbes that are relevant to broiler health. Right now, the figures are a lot of information with zero indication that any of it is useful or statistically significant, and can be deleted completely.

Author Response

Response to Reviewer 3 Comments

We thank the Reviewer for the interest in our work and for their helpful comments, which will greatly improve the manuscript. We have done our best to respond to all the points raised in the review. We would like to apologize for all oversights and for presenting the results in unclear way. Below present our answers to the Reviewer comments.

The findings are interesting and the sample size is excellent. However, the results and methods sections need work.

  1. It is not clear how you determined the statistical significance of differences between the microbiomes. No mention of your statistical methods is made in the methods. A section 2.7 or another paragraph for section 2.6 is needed for this.

Answer:

The Authors want to thank the Reviewer for this comment. By mistake the Authors forgot to add information about statistical model. The following paragraph was added to the Material and Methods section”

“2.7. Statistical Analysis

The experiments had a completely randomized design. The sample obtained by pooling digesta from 2 birds was defined as experimental unit, i.e., n=5 per treatment (10 birds per group, 1 randomly chosen chick from each pen, 2 birds pooled per sample).

Bioinformatic analysis was carried out using the R program and using the phyloseq, vegan and factoextra packages, while charts were generated using the gglpot2 and ggbiplot packages [31]. The beta diversity measure was calculated based on the Bray-Curtis method[32]. Selected sequences representing OTU were compared to the Genbank database using the BLAST algorithm[33].

All data were tested for normal distributions using the Kolmogorow-Smirnov test. An analysis of variance was conducted using Bartlett’s test. The significance of differences among groups was determined with Duncan’s multiple range test at the significance level of p < 0.05. The analyses were performed using SAS software (SAS Institute Inc., Cary, NC, USA).

The following general model was used:

Yi=μ+αiij,

where Yi is the observed dependent variable, μ is the overall mean, αi is the effect of insects’ meal, and δij is the random error.”

  1. Table 2 mentions the RMSE, but this abbreviation is not defined elsewhere. What software produced this, and what does it mean?

Answer:

            The software used for statistical analyses was mentioned above, i.e., SAS software (SAS Institute Inc., Cary, NC, USA). The Authors added the explanation of RMSE as follow:

“RMSE – root square error of the mean”

The RMSE equation is: RMSE=√MSE, where MSE is mean square error.

The authors placed information as table abbreviation: “Means represent 10 birds in 5 pooled samples, 2 birds per sample (n=5); PC—positive control (salinomycin, 60 ppm); NC—negative control (no additives); TM02—(0.2% T. molitor full-fat meal); ZM02—(0.2% Z. morio full-fat meal); TM03—(0.3% T. molitor full-fat meal); ZM03—(0.3% Z. morio full-fat meal); SD – standard deviation, RMSE – root square error of the mean; The RMSE equation is: RMSE=√MSE, where MSE is mean square error, a-c means within a row with no common superscripts differ significantly (p ≤ 0.05).

.”

According to the other Reviewer opinion we added in table: SD – standard deviation

2. Figure 1 appears to be cutoff: the Y-axis starts at 82%. I also don't know if the bar graph figures are useful. They are all different sizes, arranged differently, with missing data labels or cut-off labels, etc. Plus, we don't really need information on all these families, especially since the discussion seems to focus exclusively on Lactobacilli, Bacteroides, and a few other species.

Answer:

The Authors want to thank the Reviewer for this comment. We are really sorry for mistakes in the Figure 1. The Authors changed presented figure.

Original file were also added and the size will be determined and unified by the publisher.

To make results be presented better and clearer we add results of PCA analysis, alpha and beta diversity. We hope that way of presentation will accepted by the Reviewer. The results of this data can directly present microbial diversity in cecal digesta of broiler chickens. For further statistical analysis we have chosen bacteria with the highest relative prevalence according OUT’s analysis and discussed results.

  1. An NMDS plot might be a better way to represent the differences in microbiome visually, else Table 2 is enough. Or, focus solely on the microbes that are relevant to broiler health. Right now, the figures are a lot of information with zero indication that any of it is useful or statistically significant, and can be deleted completely.

Answer:

In reference to Reviewer comments 3 and 4 we made additional statistical analysis of alpha and beta diversity of metagenomical data. The results are presented in NMDS plots (Figure 1 and 2) according Reviewer comments.

Best regards,

Agata Józefiak

Round 2

Reviewer 2 Report

I reviewed the earlier version of this manuscript and I made some comments which were not addressed. I thank the authors for providing many changes, but the important changes have not been made. 

First of all the statistical analysis has been expanded upon, but the bioinformatics part is very vague, and the analysis itself is highly Unconventional to say the least. What bioinformatics package did the authors use? QIIME, Mothur? Or something else? Also the alpha diversity of beta diversity measures are not reported. The RMSE analysis on relative abundance is highly unconventional for this type of data, if not inappropriate. These comparisons need to be made on alpha diversity indices like chao, or Simpson etc. Where are the beta diversity tests? Eg. UNIFRAC, PERMANOVA or similar?  

Also some quotes appear in the text with quotation marks, which raise some questions. 

The figures of the PCA are extremely poor quality. Please place 95% confidence interval ellipses on them so we can see if and how the groups cluster. 

Author Response

Response to Reviewer Comments

We thank the Reviewer for the interest in our work and for their helpful comments, which will greatly improve the manuscript. We have done our best to respond to all the points raised in the Review. Below you will find our answers to the comments.

  1. Comments: First of all the statistical analysis has been expanded upon, but the bioinformatics part is very vague, and the analysis itself is highly Unconventional to say the least. What bioinformatics package did the authors use? QIIME, Mothur? Or something else?

Answer:

We used the QIIME software package based on the GreenGenes v13_8 reference sequence database.

The additional description was added to the Material and Methods section 2.6. Metagenomic analysis

Line 201-2018: Bioinformatic analysis ensuring the classification of readings by species level was carried out with the QIIME software package based on the GreenGenes v13_8 reference sequence database. The analysis consisted of stages:

  1. removal of adapter sequences - cutadapt program,
  2. quality analysis of readings and removal of low-quality sequences (quality <20, minimum length 30) - cutadapt program,
  3. paired sequence connection - fastq-join algorithm,
  4. clustering based on the selected base of reference sequences - the uclust algorithm,
  5. chimer removal sequence - ChimeraSlayer algorithm,
  6. assigning taxonomy to a selected base of reference sequences - the uclust algorithm.

We've also added the following References:

USEARCH and UCLUST algorithms

Edgar,RC (2010) Search and clustering orders of magnitude faster than BLAST, Bioinformatics 26(19), 2460-2461. doi: 10.1093/bioinformatics/btq461

fastq-join

Erik Aronesty, 2011. ea-utils : “Command-line tools for processing biological sequencing data” (http://code.google.com/p/ea-utils)

Cutadapt

Marcel Martin. Cutadapt removes adapter sequences from high-throughput sequencing reads. EMBnet.journal, 17(1):10-12, May 2011. DOI: http://dx.doi.org/10.14806/ej.17.1.200

QIIME

Caporaso JG, Kuczynski J, Stombaugh J, Bittinger K, Bushman FD, Costello EK, Fierer N,

Gonzalez Pena A, Goodrich JK, Gordon JI, Huttley GA, Kelley ST, Knights D, Koenig JE,

Ley RE, Lozupone CA, McDonald D, Muegge BD, Pirrung M, Reeder J, Sevinsky JR,

Turnbaugh PJ, Walters WA, Widmann J, Yatsunenko T, Zaneveld J, Knight R. 2010.

QIIME allows analysis of high-throughput community sequencing data. Nature Methods 7(5): 335-336.

BLAST

Altschul SF, Gish W, Miller W, Myers EW, Lipman DJ. 1990. Basic local alignment search

tool. J Mol Biol 215(3):403-410.

ChimeraSlayer

Haas BJ, Gevers D, Earl AM, Feldgarden M, Ward DV, Giannoukos G, et al. 2011.

Chimeric 16S rRNA sequence formation and detection in Sanger and 454-pyrosequenced

PCR amplicons. Genome Research 21:494-504.

Greengenes

DeSantis T. Z., Hugenholtz P., Larsen N., Rojas M., Brodie E. L., Keller K., et al. (2006).

Greengenes, a chimera-checked 16S rRNA gene database and workbench compatible

with ARB. Appl. Environ. Microbiol. 72 5069–5072. 10.1128/AEM.03006-05

We added additional information

Line: 184-190 “Specific sequences 341F and 785R were used for the amplification and the libraries preparation. Table PCR reaction with Q5 Hot Start High-Fidelity 2X Master Mix available, reaction conditions were performed in accordance with the manufacturer's requirements. Sequencing took place on the MiSeq sequencer, in paired-end (PE) technology, 2x250nt, using Illumina v2 kit. Automatic initial data analysis was performed on the MiSeq apparatus using the MiSeq Reporter (MSR) v2.6 software. The analysis consisted of two stages: automatic demultiplexing of samples and generating fastq files containing raw reads.

  1. Commnents: The alpha diversity of beta diversity measures are not reported.

Answer:

We add additional files with alpha and beta diversity measures.

  1. The RMSE analysis on relative abundance is highly unconventional for this type of data, if not inappropriate.

Answer:

We agree with the Reviewer's comments, however, we used the RMSE test only to compare the most common populations of bacteria based on relative abundance data. Bioinformatics analyzes were performed first and is the basic for RMSE analysis.

We added additional information:

Line 219-221: The RMSE – root square error of the mean was calculated. The RMSE equation is: RMSE=√MSE, where MSE is mean square error.

  1. These comparisons need to be made on alpha diversity indices like chao, or Simpson etc. Where are the beta diversity tests? Eg. UNIFRAC, PERMANOVA or similar?  

The alpha_diversity were calculated for the indices Chao1, Shannon, Simpson, Inverted Simpson, observed OUT and presented in table 2.

The beta diversity measure was calculated based on the Bray-Curtis method

  1. Comments:

Also some quotes appear in the text with quotation marks, which raise some questions. 

Answer

The Authors want to thank the Reviewer for this comment. By the mistake in the text some quotes appear.

  1. The figures of the PCA are extremely poor quality. Please place 95% confidence interval ellipses on them so we can see if and how the groups cluster. 

Answer:

We would like to thank the Reviewer for help with the bioinformatics data.

The figures of the PCA are presented with 95% confidence interval ellipses on Figure 2.

It is our hope, that the present revised version can be accepted for publication in the Journal Animals.

With kind regards

Agata Józefiak